# Proteomics Profiling with SWATH-MS Quantitative Analysis of Changes in the Human Brain with HIV Infection Reveals a Differential Impact on the Frontal and Temporal Lobes

**DOI:** 10.3390/brainsci11111438

**Published:** 2021-10-28

**Authors:** Mayur Doke, Tamizhselvi Ramasamy, Vaishnavi Sundar, Jay P. McLaughlin, Thangavel Samikkannu

**Affiliations:** 1Department of Pharmaceutical Sciences, Irma Lerma Rangel College of Pharmacy, Texas A&M University, Kingsville, TX 78363, USA; mayurdoke@exchange.tamu.edu (M.D.); tselvi@tamu.edu (T.R.); vsundar@tamu.edu (V.S.); 2Department of Pharmacodynamics, College of Pharmacy, University of Florida, Gainesville, FL 32611, USA; jpmclaughlin@ufl.edu

**Keywords:** proteomics, SWATH, HIV-associated dementia, frontal lobe, human brain, LC-MS/MS, bioinformatics

## Abstract

The chronic irreversible regression of cognitive ability and memory function in human immunodeficiency virus (HIV)-associated dementia (HAND) is linked with late-stage HIV infection in the brain. The molecular-level signatures of neuroinflammation and neurodegeneration are linked with dysfunction in HAND patients. Protein expression changes and posttranslational modification are epigenetic cues for dementia and neurodegenerative disease. In this study quantitative proteome analysis was performed to comprehensively elucidate changes in protein profiles in HIV-positive (HIV+) human brains. Frontal and temporal lobes of normal and HIV+ brains were subjected to label-free liquid chromatography-tandem mass spectrometry (LC-MS/MS) analysis using the data-independent acquisition method. Comprehensive proteomic identification and quantification analysis revealed that 3294 total proteins and 251 proteins were differentially expressed in HIV+ brains; specifically, HIV+ frontal and temporal lobes had 132 and 119 differentially expressed proteins, respectively. Proteomic and bioinformatic analyses revealed protein alterations predominantly in the HIV+ frontal lobe region. The expression of GOLPH3, IMPDH2, DYNLL1, RPL11, and GPNMB proteins was significantly altered in HIV+ frontal lobes compared to that in normal brains. These proteins are associated with metabolic pathways, neurodegenerative disorders, and dementia. These proteomic-level changes may be potential biological markers and therapeutic targets to relieve the dementia-associated symptoms in individuals with HAND.

## 1. Introduction

Human immunodeficiency virus (HIV) primarily targets immune cells, but also microglia and macrophages in the brain. Despite the advancement of antiretroviral therapy, perivascular macrophages and microglial cells of the central nervous system (CNS) are major reservoirs of HIV infection, and ultimately affect astrocytes and neurons, causing the acceleration of disease progression [1]. Clinical and experimental studies have emphasized that HIV infection is directly associated with an elevated level of reactive oxygen species (ROS) [2]. HIV infection disturbs the mitochondrial metabolic functions of adenosine triphosphate (ATP) production, electron transport oxidative phosphorylation (OXPHOS), and the oxidation of metabolites in the tricarboxylic acid cycle (TCA) [3]. HIV is also known to hijack the host cellular protein regulatory network and affect cellular function. Studies have well documented the imbalance in dynamic equilibrium during posttranslational modifications of mitochondrial protein.

Posttranslational modification may impact the signaling cascade of gene expression, which has been shown to be significantly associated with depression-related behavior, dysfunction in memory formation, and dementia [4]. Moreover, posttranslational modifications have been shown to significantly impact a protein’s function, localization, and interaction with other cellular components [5]. Moreover, aberrant posttranslational modifications of proteins can substantially affect characteristic protein folding activity, which has been shown to be linked to various developmental disorders, including neurodegenerative diseases [6,7,8,9]. Further research is needed concerning the involvement of HIV infection and disease progression in posttranslational modification of proteins in the human brain, as this subject has not yet been fully elucidated. Moreover, previous studies have reported that HIV infection predominantly targets the frontal lobe of the brain compared with other brain regions in humans and in nonhuman primate models [10,11]. Collectively, improved insights about how HIV infection alters protein posttranslational modification as a mechanism promoting neurocognitive impairment and behavioral abnormalities is warranted.

In the present study, we employed dual mass spectrometry on total protein extracts obtained from autopsied human frontal and temporal lobe tissues of HIV+ patients and compared them with tissue from normal subjects to identify differentially expressed (DE) protein candidates, seeking to specifically define altered protein expression, and determine their involvement in the pathogenesis of HIV-linked neurodegenerative diseases and neurocognitive disorders. Our study is unique in that it entails the use of frontal and temporal brain tissue obtained from HIV+ individuals for detailed sequential windowed acquisition of all theoretical fragment ion mass spectrometry (SWATH-MS) proteomic analysis to elucidate epigenetic cues at the protein level.

## 2. Materials and Methods

### 2.1. Human Brain Samples

Human brain samples were received from the National Neuro AIDS Tissue Consortium (NNTC) and National Neurological AIDS Bank (NNAB) University of California (UCLA) California, CA, USA provided high-quality and well-characterized brain tissues from donors 40–60 years of age. [12]. The brain tissue samples of frontal and temporal regions from three normal and three HIV+ individuals (mild neurocognitive disorder) were subjected to dual mass spectrometry analysis followed by quantitative proteomic analysis.

### 2.2. Sample Preparation

Ion library: Brain samples of the frontal and temporal lobes from both control and HIV patients were homogenized in harsh lysis buffer (7 M urea, 2 M thiourea, 4% CHAPS in 50 mM Tris pH 8) and pooled in one tube. Two hundred micrograms of total protein were fragmented by 4–12% sodium dodecyl sulfate-polyacrylamide gel electrophoresis (SDS PAGE), stained with Coomassie Blue and cut into 12 bands. Each gel band was individually and sequentially reduced with 10 mM dithiothreitol (DTT) for 15 min at 65 °C and alkylated with 15 mM DTT for 30 min at room temperature in the dark. The remaining indole-3-acetic acid (IAA) was quenched by the addition of 10 mM DTT. Proteins in the gel were digested overnight with a mix of trypsin/LysC at 37 °C with agitation. The resulting peptides were extracted from the gel using two successive rounds of dehydration in 50% acetonitrile (ACN) + 5% formic acid (FA) and sonication. The recovered peptides were dried in nitrogen, reconstituted in 2% FA, and purified using reversed-phase solid-phase extraction (SPE). Each of the 12 fractions was analyzed on a 60 min liquid chromatography-tandem mass spectrometry (LC-MS/MS) gradient using data-dependent acquisition (DDA) on a TripleTOF 6600 (Sciex) equipped with a MicroLC 200 (Eksigent). MS/MS fragmentation from 100 m/z to 1800 m/z was set to be triggered on the 40 most intense ions identified during a survey scan for the MS1 precursors from 350 m/z to 1250 m/z, for a charge state of 2–5 and a signal >500 cps. The exclusion time was set to 30 s after two occurrences. Proteins were identified using all 12 fractions in Protein Pilot 5.5 with fixed carbamidomethyl modifications on cysteines, trypsin as the enzyme and biological modifications as a preset search feature on the Homo sapiens proteome [13].

### 2.3. LC-MS/MS Analysis

Brain samples for protein quantification were extracted using the same lysis buffer as was used for the ion library. Forty micrograms of protein for each sample were reduced with 10 mM DTT for 15 min at 65 °C, alkylated with 15 mM IAA for 30 min at room temperature in the dark and quenched with an additional 10 mM DTT. Proteins were then precipitated with eight volumes of ice-cold acetone and one volume of ice-cold methanol overnight at −80 °C. Precipitated proteins were pelleted by centrifugation for 5 min at 13,000 RMP at 4 °C. Pellets were washed three times with ice-cold methanol and resuspended in 100 µL of 0.75 M urea + 1.3 µg of trypsin/LysC in 50 mM Tris pH 8. Proteins were predigested for 2 h at 37 °C with agitation, after which another 1.3 µg of protease was added. Digestion was allowed to proceed overnight. The resulting peptides were purified using reversed-phase SPE. Samples were analyzed on a 60 min LC-MS/MS gradient using DIA on a TripleTOF 6600 (Sciex) equipped with a MicroLC 200 (Eksigent). Each sample was analyzed twice with gas-phase fractionation (GPF); the first acquisition was performed for peptides in the 350–800 m/z range, while the second acquisition was performed for peptides in the 800 m/z to 1250 m/z range. The MS1 range in each of the GPF acquisitions was fragmented into 113 windows of 4 Da, with 1 Da overlap. For both GPF acquisitions, the MS2 range was set from 100 m/z to 1800 m/z with an accumulation time of 30 ms. For each sample, both GPF runs were analyzed using the SWATH 2.0 extension of the Peakview software (Sciex) with 10 peptides per protein, 4 MS/MS transitions per peptide, 5% false discovery rate (FDR), 12.5 retention time (RT) window and 25 ppm extracted ion chromatogram (XIC) width. The reported quantification for a protein represents the sum of both GPF files for each sample. The intensity of each protein was corrected using the total signal of each sample.

### 2.4. Protein Identification and Quantification

The ion library was generated by combining the 12 fractions in Protein Pilot for protein and peptide identification. Peptide and protein quantification were performed using the Swath 2.0 module in Peakview. Data from both GPF runs were quantified and normalized individually using 10 peptides per protein and 4 MS/MS transitions per peptide. Individual GPF runs were then summed in Excel. A list of peptides with both the protein modification and a good quality score was isolated from the ion library. These peptides were then manually reviewed and integrated in Skyline. The peptide starting position and sequence are reported, as well as the area under the receiver operating characteristic curve (AUC) of this peptide in each sample.

### 2.5. Analysis of DE Proteins Using a Bioinformatics Approach

For proteome expression profile analysis, we applied the Limma-Voom R package to identify DE proteins from a given matrix of featureCounts files. We investigated DE proteins between tissue samples in the normal and HIV+ brain groups using the “model.matrix”, “lmFit”, “eBays” and “topTable” functions [14,15,16,17]. We also calculated the log2-fold change value (FC) for each protein by dividing the expression in the HIV+ brain tissue samples by the measured expression in the normal brain tissue counterparts. The statistical significance threshold was set at *p* < 0.05.

### 2.6. Data Analysis of DE Proteins

Pathway analysis of functional enrichment for the protein list was performed using the ToppFun algorithm of the ToppGene Suite [18]. The significantly altered proteins were assigned to 3 Gene Ontology (GO) categories, including biological process, cellular compartment, and molecular function, with an FDR *p*-value cutoff of 0.05 [19,20]. Moreover, the DisGeNET Curated, genome-wide association study (GWAS) and Online Mendelian Inheritance in Man (OMIM) MedGen databases were used in the ToppFun algorithm to detect protein-associated diseases with an FDR *p*-value cutoff of 0.05 [21]. In all enriched categories, the resulting *p*-values were transformed by x = −log10(*p*) and employed to assess the fold enrichment of DE proteins by Fisher’s exact test *p*-value. Kyoto Encyclopedia of Genes and Genomes (KEGG) was employed for protein pathway annotations [22,23,24]. A corrected *p* < 0.05 indicated statistical significance. Hierarchical categories were obtained based on the KEGG database. The PANTHER classification system was utilized to analyze the subcellular localization of significantly altered proteins in categories assigned to GO with an FDR *p*-value cutoff of 0.05 (http://www.pantherdb.org/; accessed on 27 August 2021) [25,26]. Pertinently, enrichplot is an R package that uses visualization methods to help researchers interpret implemented functional enrichment analyses for the process of biological term classification of significantly altered proteins. It includes and supports visualizing enrichment results obtained from DOSE [27], clusterProfiler [28], ReactomePA [29], ggplot2 [30] and meshes [31]. Both overrepresentation analysis (ORA) and gene set enrichment analysis (GSEA) were performed using the enrichplot package.

### 2.7. Statistical Analysis

We employed a 99% confidence interval, which was used to determine differences between groups. Clustering was performed with Clustvis to detect correlations between sample groups. Principal component analyses (PCAs) were performed using Clustvis and BioVinci. For bioinformatic analysis, the statistical significance threshold was set at *p* < 0.05.

## 3. Results

### 3.1. Strategy for Proteome Analysis of Normal Human Brain and HIV+ Brain Samples

Our aim was to identify protein profile changes in the frontal and temporal lobes of brain samples to identify differences between normal and HIV+ brain tissues, and to determine the perturbations in proteins that are associated with HIV infection. To decipher protein profiles and alterations caused by HIV infection in the human brain, HIV+ and normal brain tissue samples were subjected to dual mass spectrometry analysis followed by quantitative proteomic analysis. After lysis of tissue samples, proteins were digested using a mix of trypsin/LysC at 37 °C with agitation. The resulting peptides were extracted from the gel using two successive rounds of dehydration in 50% ACN + 5% FA and sonication. Peptides were analyzed on a 60-min LC-MS/MS gradient using DDA on a TripleTOF 6600 (Sciex) equipped with a MicroLC 200 (Eksigent) (Figure 1A). Each sample was analyzed twice with GPF: the first acquisition was performed for peptides in the 350–800 m/z range, while the second acquisition was performed for peptides in the 800 m/z to 1250 m/z range (Figure 1A).

### 3.2. Accurate Mapping and Quantification of Protein Changes in the Frontal and Temporal Lobes of the Brain

The ion library was generated by combining the 12 fractions in Protein Pilot for protein and peptide identification. Peptide and protein quantification were performed using the Swath 2.0 module in Peakview. Both GPF runs were quantified and normalized individually using 10 peptides per protein and 4 MS/MS transitions per peptide (Figure 1A). We performed multivariate ordination analysis to evaluate the variability among the samples. We plotted PCA plots (Figure 1B) to visualize the quantitative measure between six HIV+ samples and six normal samples for both frontal and temporal lobe regions. The first two principal components (PCs) also reflected a percentage of variance (21.2%), and the studies did not show much variation in the plot. However, the PCA plot showing significant changes in at least one condition revealed substantial variation among samples of all four groups (Figure 1B). PC1 showed more variance in 23.3% of the PCA plot with at least one significant condition than within the PCA plot including all data. Weighing the PCA plots and the limitation of working with 12 samples, we decided to include all datasets for systematic analysis.

### 3.3. Identification and Quantitation of Global Proteomes in the Frontal and Temporal Lobes of HIV+ Brains

For global proteome analysis in brain tissues, we performed dual mass spectrometry analysis followed by quantitative proteomic analysis. The data from labeled free mass spectrometry proteomic analysis were analyzed separately for each sample. Then, the results of all sample sets were combined and analyzed, resulting in the identification of a total of 3294 proteins (Appendix A (Appendix A)). Peptides and protein quantification were performed using the Swath 2.0 module in Peakview, and quantified protein values were then used to determine the difference between the two groups. Differences between the groups was assessed with statistical analysis utilizing a 99% confidence interval. Statistical analysis revealed that at least one significant change in a condition indicated that 426 proteins were significantly DE, as shown in Figure 2A. Specifically, 251 proteins were significantly altered in HIV+ frontal and temporal lobes compared to normal controls. Comparing the protein expression levels in HIV+ and normal samples, with log2fold change, 117 proteins were upregulated and 109 proteins downregulated in frontal lobe regions of the brain. Similarly, we observed that 136 proteins were upregulated and 84 proteins downregulated in temporal lobe regions of HIV+ brains compared to those of normal subjects. A Venn diagram was utilized to compare and determine significantly changed proteins that were common among four comparison groups: total significant proteins, normal frontal lobe vs HIV+ frontal lobe, normal temporal lobe vs HIV+ temporal lobe, and HIV+ frontal lobe vs HIV+ temporal lobe (Figure 2B). Venn diagram analysis revealed that RPL11 was a significantly expressed protein shared by all comparison groups. The total significant proteins, normal frontal vs HIV+ frontal lobes and normal temporal vs HIV+ temporal lobes shared 7 proteins (DC42EP4, PPIL3, TTR, FXYD1, GPRC5B, CD99, and ABLIM1), whereas there were 19 shared proteins among the total significant proteins, normal frontal vs HIV+ frontal lobes, and HIV+ frontal lobe vs. HIV+ temporal lobe proteins (H2AZ2, TPT1, MAP2K2, MAP1LC3B, AK5, ILF2, TIGAR, KPYM, SERINC1, GARS1, RBM14, GABBR2, GMDS, TMEM245, BROX, SH3GL1, LASP1, DST, and PGK1) (Figure 2B; see also Appendix A (Appendix A)). Additionally, we found that a total of 132 and 119 proteins were significantly altered in HIV+ frontal lobes and HIV+ temporal lobes, respectively (Figure 2C; see also Appendix A (Appendix A). Furthermore, we depicted the most up- and downregulated proteins between HIV+ brains and normal brains, as shown in Figure 2D. We set a criterion (<−1.5-fold and >+1.5-fold) to select the most up-and down- regulated proteins in HIV+ frontal and temporal lobes of HIV groups compared to normal frontal and temporal lobe samples. We then selected the common top 50 up and downregulated proteins present in HIV+ frontal and temporal lobes of HIV groups as compared to normal frontal and temporal lobe samples, depicting the results in the form of a heatmap (Figure 2D). In heatmap analysis, we observed that USP24, SLC27A4, IMDH2, and STXBP3 proteins were significantly downregulated in HIV+ brain tissue samples, while GPNMB, TOMM5, ATP5IF1, CD99, CD151, DYNLL1, and DARS2 proteins were significantly upregulated (Figure 2D).

### 3.4. Identification of DE Proteins in HIV+ Brains

To elucidate the role of HIV infection on the human brain, analysis of differentially expressed protein was performed between normal and HIV+ brain tissue samples. Our major goal of the study was to understand the target proteins of HIV infections and how HIV infection dysregulates protein expression, determining how this information may further affect gene expression post-translationally. To understand the protein level changes, we set criteria to identify these proteins. We acknowledged proper protein expression when reads were ≥10 in each sample, FC was ≥1.5, and *p* was ≤0.05. First, we explored protein expression in HIV+ frontal brain tissue samples compared to normal frontal brain tissue samples. Read counts from normal and HIV+ brains were converted to log2-counts-per-million (log CPM), and the mean-variance relationship was modeled with precision weights called ‘voom’ to normalize the data, as shown in Figure 3A. We observed that a total of 80 proteins were significantly altered when we compared normal frontal lobe data with the HIV+ frontal lobe data. Out of 80 proteins, 48 were significantly upregulated, while 32 were downregulated (Appendix A (Appendix A)). Figure 3B,C show the heatmap and volcano plot depicting significantly upregulated and downregulated proteins with log2fold changes and significant *p*-values, respectively. Moreover, protein expression values for GPNBM and FBXL15 in normal individuals and HIV+ individuals were compared by plotting their normalized log2fold change expression (Figure 3D). Similarly, we tried to decode the proteomic changes in HIV+ temporal lobes compared to those in normal temporal lobes. We observed that a total of 66 proteins were significantly altered when we compared normal temporal lobes with HIV+ temporal lobes. Out of 66 proteins, 43 were significantly upregulated and 23 downregulated (Appendix A (Appendix A)). Figure 4B,C show the heatmap and volcano plot depicting the significantly upregulated and downregulated proteins with log2fold changes and significant *p*-values, respectively. Moreover, protein expression values for IMPDH2 and DYNLL1 in normal individuals and HIV+ individuals were compared by plotting their normalized log2fold change expression (Figure 4D).

### 3.5. Functional Analysis of Proteins between HIV+ Brain Samples Compared to Normal Brain Samples

To explore the functional annotation of the proteins, their biological processes, cellular components, and molecular functions were compared between HIV+ and normal subject brains based on Gene Ontology (GO). The proteomic analysis results revealed a set of 426 proteins that were further used to explore GO annotation terms (from the GO consortium), and identify related biological pathways based upon the curated KEGG pathway database. The most significantly altered biological processes included metabolic processes and biological regulation, which are considered the building blocks of life. The main cellular components identified include cell organelles and extracellular compartments. Additionally, the most significant molecular functions included protein binding and catalytic activity. GO analysis to investigate the involvement of 426 proteins in various molecular and biological processes identified significant protein differences mainly involved in biological processes such as ion transport (GO: 0006811), mitochondrion organization (GO: 0007005), acetyl-CoA metabolic processes (GO: 0006637), mitochondrial ATP synthetic processes (GO: 0015986), organophosphate metabolic processes (GO: 0019637), and ATP biosynthetic and metabolic processes (GO: 0046034) (Figure 5A). Our results revealed that significantly altered proteins were mainly associated with molecular functions, such as oxidoreductase activity (GO: 0016491), adenine transmembrane transporter activity (GO: 0015207), ATPase activity (GO: 0016887) and nicotinamide adenine dinucleotide (NADH) dehydrogenase activity (GO: 0003954), as shown in Figure 5B. The most significant changes in molecular function among the 426 proteins were those directly linked to mitochondrial ATP production and oxidoreductase activity (Figure 5B). Furthermore, we investigated the localization of 426 proteins in various cellular compartments. We observed that most of the proteins were located at mitochondria; the mitochondrial membrane, and mitochondrial protein complexes (Appendix A (Appendix A)). Moreover, the subcellular localization confirmed that proteins were significantly situated at the mitochondrial nucleoid, postsynaptic tangle, neurofibrillary tangle, and neurofilaments (Appendix A (Appendix A)).

In addition, we investigated whether posttranslational modifications of proteins affect their biological function and derail the normal functioning of individual cells. Since previous clinical, pathological, and molecular research evidence has shown that abnormal protein function contributes to dysfunction of protein signaling cascades, we utilized GWAS and OMIM MedGen databases to identify diseases associated with altered proteins, as shown in Figure 5C. Analysis reveals the substantial involvement of these proteins in diseases such as neurodegenerative disorders, Parkinson’s diseases, major depression disorder, metabolic disease, and impaired cognition. These findings suggest that there were significant differences at the protein modification level between HIV+ and normal brain samples. It can be speculated that the brain tissues in HIV+ patients were greatly affected by dementia-related signatures, and underwent great functional dysregulation leading to mitochondrial dysfunction and physiological changes.

Furthermore, we performed enrichment analysis beyond the main three categories for the retrieved differentially altered proteins. GSEA was used to identify significantly altered proteins and that may have an association with disease phenotypes. The results revealed that identified proteins were significantly associated with disease signatures, including mitochondrial complex I deficiency, mitochondrial myopathies, NADH coenzyme Q reductase deficiency, neurofibrillary degeneration, and brain ischemia. Moreover, protein modifications have been linked with cytochrome c oxidase deficiency, anaplastic astrocytoma, nervous system disorder, nerve degeneration, and mitochondrial encephalopathies (Figure 6A). The most significantly changed proteins were examined for known involvement in mitochondrial-related dysfunction and neurodegenerative disorder. We identified 20 proteins, ABCA1, ACOT7, ALDH3A2, ATP2B3, CAT, CBS, COMT, DNAJC10, EEF1A2, GAP43, GCA, GGT1, GOLPH3, KIAA1217, PARK7, PRNP, TOM1L2, TTR and VCP, that were mainly linked with dementia, impaired cognition, and mental retardation, as shown in Figure 6B. Furthermore, we mapped the 426 altered proteins against the KEGG PATHWAY mapper, revealing that these proteins were mainly involved in metabolic pathways, neurodegeneration pathways and neurodegenerative diseases, such as Alzheimer’s disease, Parkinson’s disease, and Huntington disease (Figure 6C,D).

Taken together, analyses with the GSEA, GO database, GWAS and OMIM MedGen databases revealed specific protein targets that have already been shown to be closely linked with neurodegenerative and neurocognitive disorders. The proteomic analysis revealed 426 protein targets that were significantly changed in HIV+ human brain tissues. Our analysis revealed that the frontal lobe of the HIV+ brain expressed a higher number of altered proteins than the HIV+ temporal brain region.

## 4. Discussion

The bioinformatic approach utilizing GSEA and KEGG databases for examining the biological processes and molecular functions of the significantly altered 426 protein targets revealed specific significant protein candidates. The combined proteomic and bioinformatic approach analysis narrowed down 426 protein candidate list to specific protein markers, including GOLPH3, IMPDH2, DYNLL1, RPL11, and GPNMB proteins. Previous studies have already confirmed the role of these proteins in neurocognitive diseases and neurocognitive disorders. Notably, GOLPH3 is a peripheral membrane protein localized to the trans-Golgi [which may affect Golgi morphology as well as multiple cellular processes, such as vesicular trafficking, secretion, mTOR signaling, and cell survival] [32]. It is known that GOLPH3 has the ability to bind to PtdIns(4)P and MYO18A, which can act as a linkage factor between the Golgi apparatus and the actin cytoskeleton. In the case of DNA damage, DNA-PK phosphorylation of GOLPH3 increases binding to MYO18A, activating the GOLPH3 pathway, which consequently results in Golgi fragmentation and reduced trafficking [33]. Human neurodegenerative diseases, including amyotrophic lateral sclerosis, Parkinson’s disease, and Alzheimer’s disease, have shown characteristic pathophysiological factors associated with the dysregulation of the Golgi apparatus. This dysregulation and damage to Golgi morphology are shown to be similar to perturbations of the GOLPH3 pathway [33]. Mice with knockout of a Golgi-specific PI-4-kinase, PI4KIIα, which is closely linked with the GOLPH3 pathway, exhibit late-onset neurodegenerative disease [34]. This evidence suggests there is a plausible link between DNA damage and altered Golgi function in neurodegenerative diseases. Notably, an animal model study of hippocampal proteomics analysis showed that the IMPDH2 protein was significantly altered in an Alzheimer’s disease 5XFAD mouse model. These mice demonstrated a link between cognitive decline and impaired learning with altered IMPDH2 protein function and expression [35]. Moreover, Moscato et al. showed that the IMPDH2 protein was significantly dysregulated in Alzheimer’s disease-affected brain regions [36]. The DYNLL1 protein plays an important role in maintaining the spatial distribution of cytoskeletal structures and intracytoplasmic trafficking of cargo. Previous studies have demonstrated the major role of altered DYNLL1 protein in dementia, Alzheimer’s disease, aging, and other neuropsychiatric processes [37,38]. Interestingly, the Venn diagram analysis comparing HIV+ and normal brains in both frontal and temporal regions demonstrated that both groups showed alterations in the protein RPL11. RPL11 is a ribosomal protein, and previous sequencing experiment analysis studies on amyotrophic lateral sclerosis, Alzheimer’s disease, and dementia human subject samples showed remarkable alterations in the RPL11 protein [39,40,41]. Furthermore, GPNMB is a type I transmembrane glycoprotein that has been shown to play a significant role in cell differentiation, migration, inflammation/anti-inflammation, tissue regeneration, and neuroprotection. Alteration of GPNMB protein expression is closely linked with Nasu-Hakola disease (NHD), which is a rare autosomal recessive disorder characterized by progressive presenile dementia [42]. High levels of GPNMB expression in Alzheimer’s disease brains have been proposed as a potential biomarker of Alzheimer’s disease [42,43,44].

## 5. Conclusions

In summary, this systematic exploration of the proteome of the frontal and temporal lobes in HIV+ and normal human brains and comprehensive proteome analysis revealed significant changes in protein expression in HIV+ brains. Proteomic analysis of HIV+ frontal brain lobes revealed significant alterations in GOLPH3, IMPDH2, DYNLL1, RPL11, and GPNMB protein levels. Bioinformatic evaluation of differential changes in molecular functions, biological processes, cellular components, and pathways showed the altered proteins were involved in dysregulation of metabolic pathways and the promotion of neurodegeneration. Importantly, specific changes in global protein expression levels may constitute potential markers of pathology and help guide the development of treatments for HIV patient health care.

## Figures and Tables

**Figure 1 brainsci-11-01438-f001:**
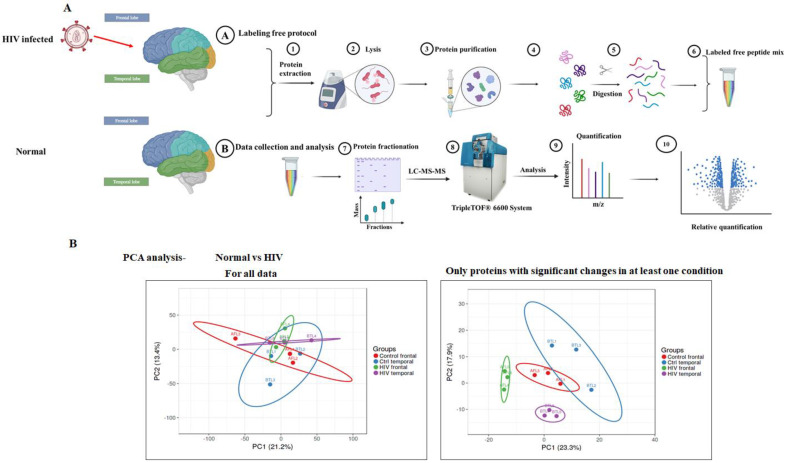
Overview of the sample preparation, analysis, and SWATH-MS analysis of brain tissue samples. (**A**) Study pipeline and proteomic workflow for proteomic analysis of frontal and temporal brain tissues from normal and HIV+ participants for data analysis. (**B**) Principal component analysis (PCA) for all samples performed for all data and only genes with significant changes in at least one condition. Principal component 1 (PC1) and principal component 2 (PC2) were identified by variance in the SWATH-MS Peakview data. The percentage of variance indicates how much variance was explained by PC1 and PC2. (Red: control frontal group; Green: HIV+ frontal group; Blue: control temporal group, violet: HIV+ temporal group).

**Figure 2 brainsci-11-01438-f002:**
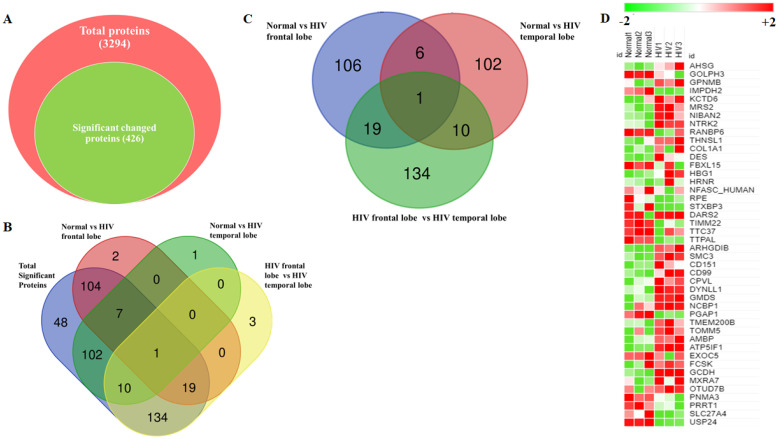
(**A**) The Venn diagram represents a comparison of significantly differentially expressed proteins in HIV+ brain tissue compared to normal groups. A total of 426 proteins were significantly expressed out of 3294 total proteins identified by proteomic analysis. (**B**) The Venn diagram used to compare and determine shared proteins with significant changes among four comparison groups: total significant proteins, normal frontal vs HIV+ frontal proteins, normal temporal vs HIV+ temporal proteins, and HIV+ frontal vs HIV+ temporal proteins. (**C**) The Venn diagram used to compare and determine shared proteins with significant changes between normal frontal vs HIV+ frontal lobes, normal temporal vs HIV+ temporal lobes, and HIV+ frontal vs HIV+ temporal lobes. (**D**) The heatmaps represent the top 50 up and downregulated proteins in HIV+ groups compared with normal groups for both frontal and temporal brain regions.

**Figure 3 brainsci-11-01438-f003:**
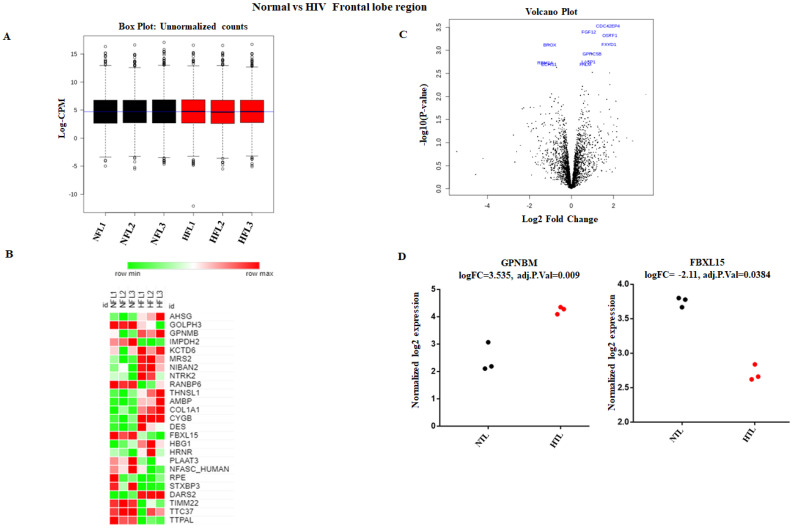
Identification of proteins differentially expressed in normal vs HIV+ frontal lobe brain tissue comparisons. (**A**) This box-whisker plot depicts the unnormalized counts of normal vs HIV+ groups, which were then transformed to normalized counts for both groups. (**B**) The heatmap represents the top up and downregulated proteins with log2fold changes. Green represents downregulation, while red represents upregulation. (**C**) Volcano plot presenting the differential expression of proteins. The *X*-axis corresponds to fold changes of −2 (downregulation) and +2 (upregulation). The *Y*-axis represents a −log10 of *p*-values. The statistically significant differentially expressed proteins are depicted in blue. (**D**) Strip chart showing significantly expressed proteins with their individual expression values in each sample. The expression of FBXL15 was downregulated in HIV+ frontal lobe tissue compared with that in normal tissue (*p* < 0.05). GPNBM was upregulated in HIV+ frontal lobe tissue.

**Figure 4 brainsci-11-01438-f004:**
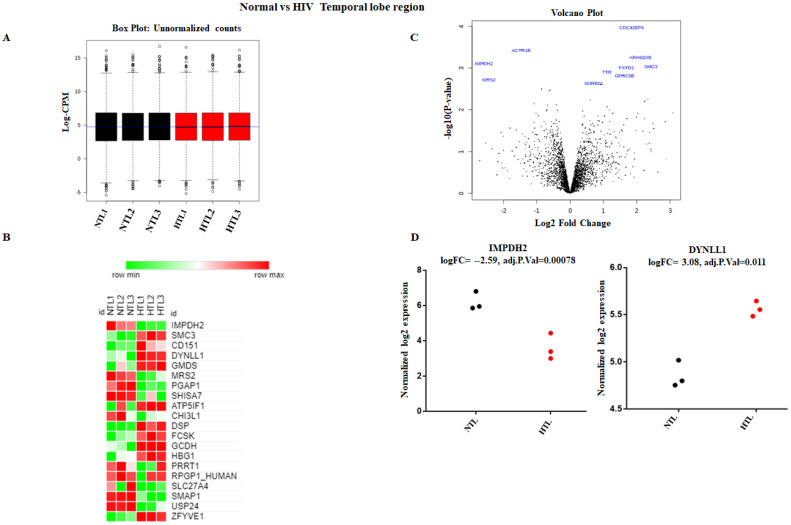
Identification of proteins differentially expressed in normal vs HIV+ temporal lobe brain tissue comparisons. (**A**) This box-whisker plot shows the unnormalized counts of normal vs HIV+ groups, which were then transformed to normalized counts for both groups. (**B**) The heatmap represents the top up- and downregulated proteins with log2fold changes. Green represents downregulation, while red represents upregulation. (**C**) Volcano plot presenting the differential expression of proteins. The *X*-axis corresponds to fold changes of −2 (downregulation) and +2 (upregulation). The *Y*-axis represents a −log10 of *p*-values. The statistically significant differentially expressed proteins are depicted in blue. (**D**) Strip chart showing significantly differentially expressed proteins with their individual expression values in each sample. The expression of IMPDH2 was downregulated in the HIV+ temporal lobe tissue compared with that in normal tissue (*p* < 0.05). DYNLL1 was upregulated in HIV+ temporal lobe tissue.

**Figure 5 brainsci-11-01438-f005:**
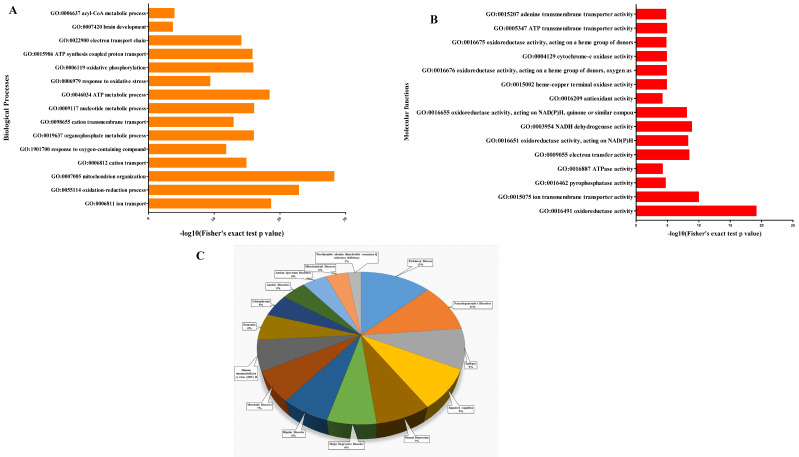
Functional analysis of protein. (**A**) Biological process identification was carried out by the PANTHER classification system, and Gene Ontology (GO) terms were retrieved. The top 15 significantly (*p* ≤ 0.05) enriched GO terms in the biological process category are shown. (**B**) Molecular function identification was carried out by the PANTHER classification system, and Gene Ontology (GO) terms were retrieved. The top 15 significantly (*p* ≤ 0.05) enriched GO terms in molecular functions are shown. (**C**) OMIM MedGen databases were used to identify the most significant (*p* ≤ 0.05) diseases associated with altered proteins and are shown in a circular chart.

**Figure 6 brainsci-11-01438-f006:**
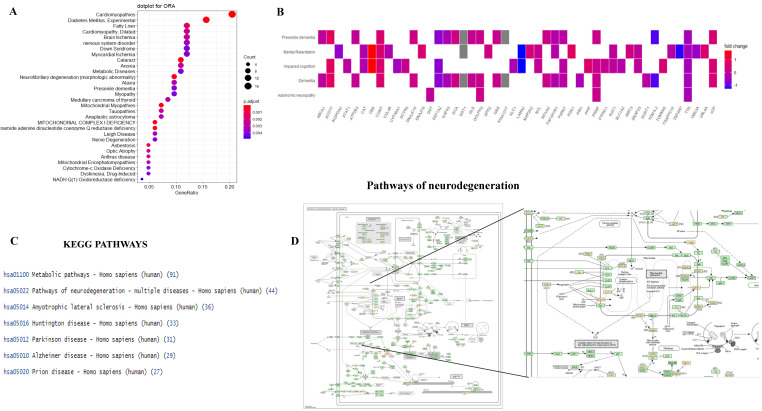
(**A**) Overrepresentation analysis (ORA) plot represents Kyoto Encyclopedia of Genes and Genomes (KEGG) pathway analysis of enriched biological terms associated with proteins. (**B**) The heat plot depicts and identified expression patterns. The relationships between proteins and disorders are displayed as a heatmap. (**C**) KEGG mapper pathway analysis of 426 protein targets identified major pathways in which these proteins are involved. (**D**)The diagram shows that the neurodegeneration pathway is one of the pathways in which the 426 protein targets are involved.

## Data Availability

The authors confirm that the data supporting the findings of this study are available within the article and/or its Appendix A.

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
