# Peer review of "Proteomics Profiling with SWATH-MS Quantitative Analysis of Changes in the Human Brain with HIV Infection Reveals a Differential Impact on the Frontal and Temporal Lobes"

_brainsci, 2021, doi:10.3390/brainsci11111438_

Round 1

Reviewer 1 Report

The authors have done an excellent work In identifying proteomic signatures between different neurological stages in HIV disease. The paper is well written. Overall, I have a  minor concern. The authors have not mentioned any kind of demographics for the samples they used in the study. It is important to provide that information.

Author Response

Rev#1. The authors have done an excellent work In identifying proteomic signatures between different neurological stages in HIV disease. The paper is well written. Overall, I have a  minor concern. The authors have not mentioned any kind of demographics for the samples they used in the study. It is important

Answer: Thank you for the suggestion. We have added the samples information and ethical statements.

Reviewer 2 Report

HIV infection during late stages of the HIV disease reduce the cognitive ability and memory function in humans, but causative connections between HIV-induced changes in host gene expression and these calamities are incomplete. In the current manuscript, Samikkannu and colleagues report on a quantitative proteomics analysis of the changes in protein content in the frontal and temporal lobes of specimens in tissue banks of HIV infected individuals. Their mass spectral analysis has allowed them to identify several proteins whose abundance is significantly different in the frontal and temporal lobes, the former of which is particularly affected by HIV infection. Using public databases, the have shown that these protein changes can be associated with GO terms characteristic of metabolic changes. The experiments and analysis are performed at a high-quality level and represent significant additions to the current characterization of HIV infections in brain tissue. In fact, I have no criticisms of the work, but I do have some comments on the preparation of the manuscript.

  1. Something is missing in lines 85-89, lines 221-223, and the legend to Fig 3C
  2. Figure 2D: I assume that this are the most differentially expressed genes. Please specify, including the method used to identify the genes.
  3. Please specify the lfc values at the extremes of the scales for heat plots in Figs 2D, 3B, and 4B.
  4. I see no red and green points in Volcano plots
  5. The labels on the figures are illegible. Perhaps the figure in the pdf file are drafts only? Please make sure that the quality of the figures is improved. A larger font may be helpful
  6. Why are the heat maps called “bar plots” in the figure legends?
  7. Please provide links to the databases used for the analysis (PANTHER, etc).

Author Response

Rev#2. HIV infection during late stages of the HIV disease reduce the cognitive ability and memory function in humans, but causative connections between HIV-induced changes in host gene expression and these calamities are incomplete. In the current manuscript, Samikkannu and colleagues report on a quantitative proteomics analysis of the changes in protein content in the frontal and temporal lobes of specimens in tissue banks of HIV infected individuals. Their mass spectral analysis has allowed them to identify several proteins whose abundance is significantly different in the frontal and temporal lobes, the former of which is particularly affected by HIV infection. Using public databases, the have shown that these protein changes can be associated with GO terms characteristic of metabolic changes. The experiments and analysis are performed at a high-quality level and represent significant additions to the current characterization of HIV infections in brain tissue. In fact, I have no criticisms of the work, but I do have some comments on the preparation of the manuscript.

  1. Something is missing in lines 85-89, lines 221-223, and the legend to Fig 3C

Answer- Thank you for pointing out our mistake. We have checked and updated the above-mentioned lines

  1. Figure 2D: I assume that this are the most differentially expressed genes. Please specify, including the method used to identify the genes.

Answer- We have now mentioned the method which we used to identify significantly expressed protein in HIV vs normal samples. Please check the page no.6 and line no. 241-245.

  1. Please specify the lfc values at the extremes of the scales for heat plots in Figs 2D, 3B, and 4B.

Answer- We have included lfc values in Figs 2D, 3B, and 4B and updated figures 2,3 and 4.

  1. I see no red and green points in Volcano plots

Answer- Thank you for noticing our mistakes. Please find updated figure legends for figure 3 and 4. We have mentioned as “The DE proteins with statistical significance depicted in blue color.”

  1. The labels on the figures are illegible. Perhaps the figure in the pdf file are drafts only? Please make sure that the quality of the figures is improved. A larger font may be helpful

Answer- We have uploaded higher resolution images to journal system.

  1. Why are the heat maps called “bar plots” in the figure legends?

Answer- We have changed the word from  “bar plots” to “ heatmaps” in the figure legends.

  1. Please provide links to the databases used for the analysis (PANTHER, etc).

Answer- Thank you for the suggestion. We have added the link in the methodology section. Please check Page no. 4 and line no. 158.